# The Antioxidant Activity of Wild-Growing Plants Containing Phenolic Compounds in Latvia

**DOI:** 10.3390/plants12244108

**Published:** 2023-12-08

**Authors:** Renāte Teterovska, Inga Sile, Artūrs Paulausks, Liga Kovalcuka, Rudīte Koka, Baiba Mauriņa, Dace Bandere

**Affiliations:** 1Department of Pharmaceutical Chemistry, Riga Stradiņš University, LV-1007 Riga, Latvia; dace.bandere@rsu.lv; 2Department of Pharmaceuticals, Red Cross Medical College of Riga Stradiņš University, LV-1009 Riga, Latvia; 3Department of Applied Pharmacy, Riga Stradinš University, 16 Dzirciema Street, LV-1007 Riga, Latvia; inga.sile@rsu.lv (I.S.); baiba.maurina@rsu.lv (B.M.); 4Latvian Institute of Organic Synthesis, 21 Aizkraukles Street, LV-1006 Riga, Latvia; 5Laboratory of Finished Dosage Forms, Riga Stradiņš University, 16 Dzirciema Street, LV-1007 Riga, Latvia; arturs.paulausks@rsu.lv; 6Clinical Institute, Faculty of Veterinary Medicine, Latvia University of Life Sciences and Technologies, LV-3004 Jelgava, Latvia; liga.kovalcuka@lbtu.lv; 7Department of Biology and Microbiology, Riga Stradinš University, 16 Dzirciema Street, LV-1007 Riga, Latvia; rudite.koka@rsu.lv; 8Baltic Biomaterials Centre of Excellence, Headquarters at Riga Technical University, LV-1658 Riga, Latvia

**Keywords:** antioxidants, phenolic compounds, phenolic acids, flavonoids, tannins, herbal extracts

## Abstract

Ethnobotanical reports from Latvia show that *Tanacetum vulgare*, *Calluna vulgaris*, *Quercus robur*, *Artemisa absinthium*, and *Artemisia vulgaris* contain phenolic compounds that have antioxidant properties, which can be beneficial in the treatment and prophylaxis of many diseases. The aim of this study was to characterize the phenolic compounds and antioxidant properties of these plants. Plant extracts were prepared using ethanol or acetone and then freeze-dried. Their total phenolic content (TPC), total flavonoid content (TFC), and total tannin content (TTC) were determined and characterized by HPLC. Their antioxidant properties were determined using a DPPH (2,2-diphenyl-1-picrylhydrazyl) radical scavenging assay. *C. vulgaris* herb and *T. vulgare* leaf extracts contained the highest amounts of flavonoids, but the bark of *Q. robur* had mostly tannins and phenolic acids. *A. absinthium* and *A. vulgaris* had the lowest amounts of polyphenols. When compared using extraction solvents, all acetone extracts had more TPC, more TFC, and better antioxidant activity. All plants contained chlorogenic acid, which contributes to antioxidant properties. The analysed plant extracts could be used in future studies to develop medicinal products with antioxidant properties.

## 1. Introduction

Polyphenolic compounds are secondary metabolites that occur naturally in plants. More than 10,000 different polyphenols are found in foods (fruits and vegetables) and medicinal plants [1,2]. Polyphenols, mainly flavonoids, tannins, and phenolic acids, are known to have positive effects on human health [2]. Many studies show that these health benefits are related to polyphenols’ antioxidant and anti-inflammatory properties [1,3,4].

An important group of polyphenols with antioxidant properties are flavonoids. Flavonoids are the largest group of polyphenols and are divided into several subclasses based on their structure (anthocyanins, chalcones, flavanols or catechins, flavanones, flavones, and isoflavones). Most of them are bound to sugars that form beta-glucosides [2]. More polar constituents such as flavonoid glycosides and aglycones are water- and alcohol-soluble, but fewer polar substances such as isoflavones, flavanones, methylated flavones, and flavanols require organic solvents (chloroform, diethyl ether, dichloromethane, and ethyl acetate). Flavan-3-ol structure-based compounds can be extracted from plants using water [4].

Okuda’s team [5,6,7] has carried out extensive research on tannins, their structures, and their pharmacological properties over the years. These compounds are biosynthesized in plants through a series of enzymatic reactions via the shikimic acid pathway. Their publications note the anti-inflammatory, antioxidant, anticancer, antiviral, and other medicinal properties of hydrolysable and condensed tannins and their metabolites [5,6,7]. The hydrolysable tannin gallic acid (3,4,5-trihydroxybenzoic acid) takes the form of colourless to yellow crystals that dissolve in water, ethanol, acetone, ether, and glycerol [8,9]. Gallic acid has several medicinal properties mentioned in the literature, such as antibacterial [8], anti-inflammatory, antioxidant [10,11,12], and neuroprotective [13] properties, and it protects the colon against issues induced by oxidative stress [14]. Ellagic acid is derived from gallic acid. It is an odourless powder that presents as cream-colored needles (from pyridine) or a yellow powder that is slightly solvable in water and ethanol and soluble in alkalies but insolvable in ether [15]. Some medicinal uses of ellagic acid are based on its anti-inflammatory, antioxidant, and antiproliferative properties [12,15,16].

Many phenolic acids have antioxidant properties. Typically, they are divided into two subgroups: hydroxybenzoic and hydroxycinnamic acids. Chlorogenic acid and its derivates are widely distributed soluble phenolic acids in plants [17]. Chlorogenic acid (3-(3,4-dihydroxycinnamoyl) quinic acid) is the cinnamate ester of caffeic acid and quinic acid [18]. Overviews of chlorogenic acid describe its various applications in many industries, such as chemical, food, health care, and cosmetics [19]. Chlorogenic acid is a well-known antioxidant [20].

The antioxidant activities of tannin derivatives and phenolic acids depend on the number of hydroxyl moieties attached to the aromatic rings of cinnamic acid or benzoic acid molecules; therefore, hydroxylated cinnamates present more effectiveness than their benzoate counterparts. In the case of flavonoids, free radical scavenging activities depend on the *o*-dihydroxy structure in the B ring, the 2,3, the double bond in conjugation with the 4-oxo function in the C ring, and the 3- and 5-OH groups in the A and C rings [17,21].

In our work, we chose to study the flower and leaf of *Tanacetum vulgare* L. (*T. vulgare*), or common tansy, the herbs *Artemisia absinthium* L. (*A. absinthium*), or wormwood, *Artemisia vulgaris* L. (*A. vulgaris*), or common mugwort, and *Calluna vulgaris* (L.) *Hull* (*C. vulgaris*), or heather, and the bark of *Quercus robur* L. (*Q. robur*), or English oak, growing in Latvia. These plants are known to have phenolic compounds and have been used in traditional and folk medicine in Europe and have mentions in Latvian folk medicine. The Asteraceae and Ericaceae families are amongst the most popular plant families used in Latvian folk medicine. *A. vulgare*, *A. absinthium*, and *T. vulgare*, which belong to the Asteraceae family, have varied uses in the treatment of digestive, neurological, skin, and respiratory system diseases, as described in folklore. *C. vulgaris* from the Ericaceae family has been used for pain and inflammation. *Q. robur* and *A. absinthium* were amongst the most frequently mentioned medicinal plants in folk literature [22]. Furthermore, our earlier literature review showed the potential anthelmintic effects of these selected plants [23]. Additionally, experimental data have shown the presence of polyphenols in and the antibacterial activities [24,25] of extracts of *C. vulgaris*, *Q. robur*, and *T. vulgare*. Antioxidant effects could be beneficial in treating infectious diseases and encouraging healing [26]. Thus, we aimed to characterize the phenolic compounds and antioxidant activity of said plants’ extracts using a free radical scavenging assay.

## 2. Results

### 2.1. Phenolic Compound Characterization

#### 2.1.1. Determination of Total Phenolic Content in Acetone and Ethanol Extracts

Two solvents (acetone and ethanol) were used to prepare extracts; then, the solvents were removed, and the lyophilized extracts were compared. The highest content of phenolic compounds was observed in *Q. robur* bark extracts (301–316 mg GEA/1 g wt), followed by *C. vulgaris* herb extracts (286–294 mg GEA/1 g wt). The lowest total phenolic content (TPC) was found in the extracts of the *A. absinthium* herb. The extracts of *T. vulgare* leaf had more TPC than *T. vulgare* flower extracts. Acetone extracts showed higher yields of phenolic compounds than ethanol extracts. The results are presented in Table 1.

#### 2.1.2. Determination of Total Flavonoid Content

The total flavonoid content (TFC) of the plant extracts is shown in Table 1. *C. vulgaris* herb and *T. vulgare* leaf extracts contained the highest amounts of flavonoids (46.14–55.08 mg QE/1 g wt), and the acetone extracts of these plants also had high amounts. Although the extract of *Q. robur* bark contained the highest amount of TPC, it had a minimal amount of TFC (less than 6 mg QE/1 g wt). Additionally, TFC was low in the *A. absinthium* and *A. vulgaris* herb extracts (3–16 mg QE/1 g wt).

#### 2.1.3. Determination of Tannin Contents in Herbal Decoctions

Figure 1 shows the total tannin contents. The *A. vulgaris* herb had the highest tannin yield of 3.02% (SD ± 0.67). The *Q. robur* bark had 2.95% (SD ± 0.56), but the *C. vulgaris* herb and *T. vulgare* leaf had approximately equal amounts of tannins at 2.45% (SD ± 0.17) and 2.43% (SD ± 0.37), respectively. Fewer tannins were found in the *T. vulgare* flower, 2.01% (SD ± 0.18), and the least were found in the *A. absinthium* herb, 1.10% (SD ± 0.07).

#### 2.1.4. Phenolic Compound Characterisation via High-Performance Liquid Chromatography (HPLC)

Figure 2 shows spectra of the plant decoctions. The standard solution of gallic acid showed a peak at 1.7 min, but the chlorogenic acid standard solution showed a peak at 7.3 min. An ellagic acid peak was registered at 14.5 min. Based on standard peaks, the plant decoctions’ chromatograms were analysed. Gallic acid and chlorogenic acids were found in the *C. vulgaris* herb, the *Q. robur* bark, and the *A. absinthium* herb, but the *A. vulgaris* herb only presented chlorogenic acid. The phenolic acids found in the *T. vulgare* flower were gallic and chlorogenic acid, but in the *T. vulgare* leaf, only chlorogenic acid was separated. None of the decoctions had peaks at 14.5 min; therefore, they do not contain ellagic acid.

### 2.2. Determination of Antioxidant Activity Using a DPPH (2,2-Diphenyl-1-picrylhydrazyl) Radical Scavenging Assay

The radical scavenging activity of the antioxidants of all the extracts is shown in Table 1. The acetone and ethanol extracts of *Q. robur* bark showed the best radical scavenging activity (IC_50_ 83.95 and 96.16 µg/mL, respectively). In all, the acetone extracts had better antioxidant activity. The radical scavenging activity of the extracts is in the following order from the lowest to highest activity levels: *A. absinthium* herb extract < *A. vulgaris* ethanol extract < *T. vulgare* flower ethanol extract < *T. vulgare* leaf ethanol extract < *T. vulgare* flower acetone extract < *A. absinthium* acetone extract < *T. vulgare* leaf acetone extract < *C. vulgaris* herb ethanol extract < *C. vulgaris* herb acetone extract.

When compared using extraction solvents, all the acetone extracts had more TPC, more TFC, and better antioxidant activity than the ethanol extracts. The amount of TPC showed a strong correlation with radical scavenging activity (r = −0.986, *p* = 0.000, *p* < 0.01) in all the extracts. If compared on the basis of the type of solvent used in the preparation of the extract, both acetone and ethanol extracts showed a strong correlation between TPC and radical scavenging activity (r = 1.00, *p* < 0.01), whereas TFC and radical scavenging activity displayed no correlation (r = −0.37, *p* = 0.468 acetone extracts, r = −0.429, *p* = 0.37 ethanol extracts). Both the *Q. robur* bark and *C. vulgaris* extracts had the highest radical scavenging activity and TPC amount. Although the *Q. robur* bark had a low amount of TFC, the other phenolic compounds were assumed to account for the radical scavenging activity in this case.

## 3. Discussion

All investigated plants contained polyphenols and had varied radical scavenging activity levels. Although the total phenolic and total flavonoid contents have been previously reported for the plants involved in our study, less is known about their tannin contents.

*Q. robur* bark had a higher TPC (301 mg GEA/1 g wt ethanol extract and 316 mg GEA/1 g wt acetone extract) than in an earlier study. Ethanol/water extracts of *Q. robur* bark had up to 79 mg GEA/1 g wt [27]. However, in our study, *Q. robur* bark’s TFC values, 5.11 mg/g ethanol extract and 6.20 mg/g acetone extract, were significantly lower than in another data study, which reported 72–73 mg/g in an ethanol/water extract and 35 mg in a water extract [27]. The ethanol and acetone extracts of the *Q. robur* bark in our study most likely had constituents belonging to tannin groups. *Q. robur* is a well-known source of tannins. On average, its bark contains up to 16% tannins [28,29]. According to Eur. Pharm. Monograph, Oak bark (*Quercus cortex*) should have at least 3% tannins [30]; our data show that the *Q. robur* bark had 2.95% SD = ±0.56, which barely meets the minimum amount required. Organic polar solvents such as methanol, acetone, and ethanol could be better for the extraction of tannins than water. Additionally, location and growing conditions also affect the amounts of polyphenols in *Q. robur* bark [29].

Extracts of *C. vulgaris* herb were rich in polyphenol compounds (TPC, 286–294 mg GEA/1 g wt). Earlier studies on *C. vulgaris* herb showed that the total tannins in its extracts can range widely (from 22 to 84 g of tannic acid equivalents (TAEs)/kg dried mass), and it has a diverse phenolic profile depending on its location and season [31,32].

*A. vulgaris* has a diverse phytochemical profile. The decoction of *A. vulgaris* contained 3% tannins. *A. vulgaris* herb [33] has been reported to have tannins but in unspecified amounts. Previously reported flavonoids include apigenin; derivatives of kaempferol and quercetin; coumarin compounds, such as esculin, umbelliferon, and scopoletin; and phenolic acids such as caffeic acid and quinic acid and their derivatives [33]. *A. absinthium* leaf has been reported to have caffeic acid, p-coumaric acid, m-coumaric acid, o-coumaric acid, ferulic acid, gallic acid, vanillic acid, salicylic acid, protocatechuic acid, and various flavonoids [34]. In our study, a water decoction of *A. absinthium* herb was used, which could account for a different phenolic acid that was discovered—chlorogenic acid.

There are many studies on *T. vulgare* flower and its essential oil, but its leaf alone has been less frequently analysed [35,36]. Therefore, the authors wanted to compare the phenolic composition and antioxidant activities of two parts of the plant—both the flower and leaf. It was previously reported that ethanol extracts of *T. vulgare* flower have phenolic acids, such as chlorogenic acid, and trace amounts of caffeic acid [37], ferulic acid derivates, hydroxycinnamoylquinic acids, and flavonoids and their glycosides [38,39]. On the other hand, *T. vulgare* leaf has been reported to feature chlorogenic acid and its derivatives, caffeic acid, and hydroxycinnamic acid derivatives. Its amount of chlorogenic acid was especially high [40]. Similarly, both of our *T. vulgare* decoctions contained chlorogenic acid, which was previously reported to be the most abundant phenolic in *T. vulgare* plants [40]. The methanol extract of the aerial parts of *T. vulgare* had luteolin, axillarin, and 3,5-O-dicaffeoylquinic acid. All identified phenolics were associated with antioxidant activity, but the strongest was 3,5-O-dicaffeoylquinic acid, IC_50_ = 9.7 μM [41]. However, our results are more in agreement with the findings of Ivanscu et al. [42]. Their extract of *T. vulgare* aerial parts was rich in chlorogenic acid and had good antioxidant activity. All extracts had some antioxidant activity, yet their phenolic profiles were different.

Although Folin–Ciocalteu reagent is not selective for polyphenols or phenolics, it is often used to determine the contents of phenolic compounds [43]. In addition, the total phenolic content has been associated with antioxidant activity [44]. The correlation of polyphenols and the antiradical activity of our plant extracts is in agreement with previously reported studies [44,45,46]. Although the correlation of TPC with DPPH is well known, it is more complicated to correlate a single tannin with antioxidant activity. Tannins are secondary antioxidants; as they do not directly donate H+, their antioxidant ability is related to radical scavenging, the ability to consume dissolved oxygen, to chelate metals (Fe II), and the reduction of Fe III. Diverse structures of tannins complicate the analysis of their activities as the contents of tannins and polyphenols are usually expressed as reference substances (catechin for condensed tannins and gallic acid for hydrolysable). At the same time, in extracts, they occur as different molecules [45,47]. The reduction of tannins in extracts decreased the amount of other phenolic compounds and antiradical activity [46]. Similar observations are consistent for our plant extracts, extracts with a higher content of polyphenols had better antiradical activity. The extracts had a strong correlation between TPC and antiradical activity, but their flavonoid content was low, leading to the idea that tannins and phenolic acids are responsible. Quercus species are well known for their tannins [48], but it is important to determine the amounts and types of tannins and their relationship to the antiradical activities of herbaceous plants. Therefore, ethnobotanical testimonies can be a good source of information, which leads to our plant choices.

The 50% ethanol extract of the *T. vulgare* leaf reported by Sowa [40] had an antioxidant activity level of DPPH 257.17 μmol TE (Trolox equivalents)/g. Additionally, TPC and antioxidant activity were reported to be positively correlated for *T. vulgare* leaf [40]. This corresponds to our findings. *C. vulgaris* has many phenolic compounds with antioxidant activity, which could explain the high activity of its extracts [31,49]. Even *C. vulgaris* honey has antioxidant activities [50]. Chlorogenic acid was reported as an antioxidant in the *C. vulgaris* herb [49].

As chlorogenic acid was found in all the plant decoctions in our study, it could be responsible for the antioxidant activity of our extracts, although in many cases, its mechanisms of action, bioavailability, and toxicology are unclear [51,52]. In addition, our investigated decoctions (apart from *A. vulgaris* herb and *T. vulgare* leaf) had gallic acid. Gallic acid and its derivatives have been reported to have versatile antioxidant properties. It has anticancer and cardiovascular protective properties through antioxidant activity [53].

None of the plant decoctions had ellagic acid. Although ellagic acid is found in *Quercus* family plants, we did not find it in the *Q. robur* bark decoction. Earlier reports show that Oak bark yields a mixture of condensed tannins and ellagitannins which, upon hydrolysis, form ellagic acid. Greater contents of ellagic acid are typically found in the fruits of plants, but it can be found in all plant parts [16].

A 2008 study [54] reported that water extracts from *A. vulgaris* herb had DPPH radicals at IC_50_ = 11.4 μg/mL, but a later study used methanol and ethanol for extracts and presented their results as radical scavenging activity and Trolox equivalent antioxidant capacity. The 70% ethanol extract of *A. vulgaris* had a DPPH of IC_50_ 0.976 μg/mL, which is higher than our results [55]. The antioxidant activities of *A. vulgaris* differ for various plant parts, and various solvents have been used in studies.

*Q. robur* bark had a relatively high free radical scavenging activity level of 96 µg/mL for the ethanol extract and 83.9 µg/mL for the acetone extract in our study. DPPH antioxidant activity was also reported in studies on bark of *Q. robur* growing in Poland [27] and Spain [56]. In a Polish study, 60% ethanol extracts had the best antioxidant activity, TPC, and gallic acid yield [27], but Galinanes’s (Spain) study reported that 2% Na_2_SO_3_ was more active than a water extract [56]. Moreover, that study reported that specific phenolic compounds and activities depend on the extraction solvent used, which corresponds with our results.

The same observations were noted for the *A. absinthium* extracts. The study re-proved that the antioxidant activity of *A. absinthium* herb, assessed via DPPH, depends on the extraction solvent used (methanol extract: 6.77 µg/mL, water extract: 7.37 µg/mL, and ethyl acetate extract: 41.45 µg/mL) [34]. The methanolic extract of *A. absinthium* showed antioxidant activity of up to 9.38 mg/mL [57]. Furthermore, it was reported that *A. absinthium* extracts obtained using more polar solvents (ethyl acetate and methanol) tend to have better antioxidant activity than those obtained using nonpolar solvents (chloroform and petroleum ether). Additionally, a higher concentration of extraction solvent led to better antioxidant activity in the case of *A. absinthium* in that study [58]. This is in good agreement with our results, as *A. absinthium* acetone extracts showed better antioxidant activity than ethanol extracts.

Although reports on the antioxidant contents of plants have variable extract preparation methods and representations of data, some differences in antioxidant activity could be the result of variances in the polarity and pH of the solvent used for extraction. In addition, it was reported that *C. vulgaris* herb chemical profiles and antioxidant activities are influenced by environment, fertilisation and harvest seasons, and vegetation stage. The polyphenol amount increased in higher altitudes [59], and the phenolic composition was enriched during the summer. Seasonal changes and harvest location should be considered when comparing studies [31,32,60,61]. It is possible that the same principles can be applied to other herbs.

The DPPH radical scavenging assay is a commonly used method of measuring antioxidant efficiency, although it has limits in sensitivity. Nevertheless, this assay offers advantages, including precision, reproducibility, and simplicity, rendering it a cost-effective methodology [62]. The DPPH radical scavenging assay is not the only method used to determine the potential antioxidant activity of herbal material. Various assays can be applied to measure antioxidant activities, such as spectrometric (TRAP—Total Peroxyl Radical Trapping Antioxidant Parameter; CUPRAC—Cupric Reducing Antioxidant Power; FRAP—Ferric Reducing Antioxidant Power; PFRAP—potassium ferricyanide reducing power, etc.), chromatographic (gas or high-performance liquid), and electrochemical techniques [63]. It should be noted that the combination of several methods is advantageous as it provides a better understanding of extracts’ antioxidant ability.

Tannins have been considered antinutritional for many years, but recent studies show their health benefits. To evaluate tannin’s effect on health, several factors must be considered such as animal species, types of tannins, and concentrations in plants or feed [64,65]. A review study reported that tannins have antibacterial effects, improve the intestinal microbial ecosystem, and enhance the growth and well-being of poultry and monogastric animals if used at small levels (<5% of the dry weight of feed) [65]. Some animals, including humans, pigs, and mice, have proline-rich enzymes in their saliva that neutralize the negative effects of tannins [64,66]. In addition to the levels of tannins in feeds and herbs, the type of tannins influences health benefits. Hydrolysable tannins did not show a negative impact on animal well-being, although high doses can cause adverse effects. Condensed tannins are considered safe in moderate amounts [64,65,67]. A study of the human gut microbiome showed positive effects of condensed tannins [66]. Ruminants can benefit from tannin intake as it reduces bloating and intestinal parasites [64,68]. Tannin metabolism and its relationship with the gut microbiome is complex due to tannins’ structural diversity and high molecular mass. Hydrolysable tannins interact differently with the gut microbiome than condensed tannins, and often tannin metabolites are responsible for systemic pharmacological effects [69]. Although our study’s plants have reports on various polyphenols, including tannins, relation to antibacterial, antiparasitic, antioxidant, and anti-inflammatory effects, there are still gaps in the knowledge about precise effects and the correlation of compounds and activities. Therefore, it is necessary to monitor tannin levels in herbal medicine and determine their effects on health.

## 4. Materials and Methods

### 4.1. Plant Materials

Herbaceous plants were collected during flowering in the summer of 2019. *A. absinthium* herb and the *Q. robur* bark were collected in the Sigulda district, 57.081708, 24.780800, *T. vulgare* flower and leaf were collected in the Gulbene district, 57.169165, 26.717434, and *A. vulgaris* and *C. vulgaris* herbs were collected in the Riga district 56.872088, 24.287890. Vouchers labelled *A. Absinthium* 2019, *A. vulgaris* 2019, *C. vulgaris* 2019, *Q. robur* 2019, and *T. vulgare* 2019 are kept in the internal herb storage of Pharmaceutical Chemistry Department of Riga Stradiņš University. Plant parts were dried at a room temperature, according to general WHO guidelines [70], and stored in sealed paper boxes in shade. Herbs were ground in a coffee mill and sieved through 2 mm sieves.

### 4.2. Chemicals and Reagents

Water was distilled and purified using a Stakpure GmpH water system (Niederahr, Germany). Gallic acid, ellagic acid, AlCl_3_, and Trolox were purchased from Acros Organics (Geel, Belgium). Chlorogenic acid was purchased from the HWI group (Rülzheim, Germany). Na_2_CO_3_ was obtained from Honeywell (Charlotte, NC, USA), and 2,2-diphenyl-1-picrylhydrazyl (DPPH) was obtained from Alfa Aesar (Kandel, Germany). Folin–Ciocalteu reagent was purchased from Fisher Scientific (Loughborough, UK). All solvents used were analytical or HPLC grade. Acetonitrile and formic acid were purchased from Sigma-Aldrich (St. Louis, MO, USA).

### 4.3. Extract Preparation

The ground plant material (5 g) was macerated for 80 min in 50 mL of 50% ethanol or 50% acetone at a room temperature. After filtration, the solvent was removed via rotary vacuum evaporation and then freeze-dried via lyophilisation.

### 4.4. Determination of the Total Phenolic Content

The total phenolic content of the extracts was determined using the Folin–Ciocalteu colorimetric method described by Kähkönen [71] with slight modifications. In short, 20 µL of extract was added to a 96-well plate and mixed with 100 µL of 10% Folin–Ciocalteu reagent, followed by the addition of 80 µL of a 7.5% Na_2_CO_3_ solution. After incubation at room temperature for 30 min in the dark with slight shaking, the absorbance at 765 nm was measured on a Hidex Sense microplate reader. Gallic acid was used as a standard for the calibration curve. The total phenolic content was expressed as milligrams of gallic acid equivalent (GAE) per g of lyophilized extract. All measurements were made in triplicate.

### 4.5. Determination of the Total Flavonoid Content

The total flavonoid content (TFC) was determined using the colorimetric method described by Wang [72]. In brief, a 100 μL sample solution was mixed with the same volume of 2% aluminium trichloride in methanol. A blank was prepared similarly by adding a 100 μL sample solution to methanol without AlCl_3_. After incubation at room temperature for 10 min, the absorbance at 415 nm was measured on a Hidex Sense microplate reader. The calibration curve was prepared using various concentrations of quercetin (0–250 μg/mL) dissolved in methanol. The TFC was expressed as mg of quercetin equivalent (QE) per g of lyophilized extract. All measurements were made in triplicate.

### 4.6. Determination of Tannin Content in Herbal Decoctions

The amounts of tannins were analysed using a method provided by the *European Pharmacopoeia*, 8th edition. The decoctions were prepared from 1 g of dry herbal material and treated as was described in *Eur. Pharm 8* [73]. After 30 min of incubation in the dark, absorbance was measured at 760 nm (Mettler Toledo UV7 spectrophotometer). Results were calculated as percentages of tannins expressed as pyrogallol in dried herb. All measurements were performed in triplicate.

Tannin content % = [62.5(A_1_ − A_2_) m_2_]/[A_3_ × m_1_], where m_1_ is mass of the sample to be examined, in grams, m_2_ is the mass of pyrogallol, in grams, A_1_ is the absorbance of the total polyphenols, A_2_ is the absorbance of polyphenols not adsorbed by the hide powder, and A_3_ is the absorbance of standard pyrogallol.

### 4.7. Phenolic Compound Characterisation Analysed Using High-Performance Liquid Chromatography (HPLC)

The plant decoctions were filtered through a PTFE membrane syringe filter (0.45 μm pore size). Standard solutions were prepared using 10 mg of gallic acid, ellagic acid, and chlorogenic acid dissolved in ethanol. The analysis was performed on a Dionex UltiMate 3000 HPLC-UV (Thermo Fisher scientific, Waltham, MA, USA). The separation was carried out with a Supelco Ascentis (Supelco, Darmstadt, Germany) express C18 column (4.6 × 100 mm, 2.7 µm particle size, 90 A), using gradient elution with mobile phases A (0.1% formic acid in water) and B (acetonitrile) at a flow rate of 1 mL/min. The gradient conditions were as follows: 5–45% B, min 0–45 min; 45–5% B, 45–50 min. The column oven was set at 40 °C, and the sample injection volume was 1 µL. Peaks were detected at 360 nm UV.

### 4.8. Determination of Antioxidant Activity Using a DPPH (2,2-Diphenyl-1-picrylhydrazyl) Radical Scavenging Assay

DPPH was used to assess the free radical scavenging (antioxidant) capacity of the extracts. The DPPH radical scavenging activity was measured according to Brand-Williams [74] with some modifications. For the assay, 20 μL of extract diluted in 50% ethanol or 50% acetone was mixed with 180 μL of DPPH in methanol (40 μg/mL) in the wells of a 96-well plate. The plate was kept in the dark at room temperature for 15 min. Decreases in absorbance at 517 nm were measured using a Hidex Sense microplate reader. Ascorbic acid solutions in the concentration range of 0–800 μg/mL were used as a standard, and ethanol and acetone were used as a control. The extract was tested in a range of concentrations to establish the EC_50_ (the concentration that reduced the absorbance of DPPH by 50%). The radical scavenging activity was calculated using the following formula:

DPPH radical scavenging activity % = [(A_0_ − A_1_)/A_0_] × 100, where A_0_ is the absorbance of the control and A_1_ is the absorbance of the sample.

### 4.9. Statistical Analysis

Quantitative results are expressed as the mean ± standard error (SD) of three independent experiments. Descriptive statistical analyses (an ANOVA (a one-way analysis of variance) and a Spearman correlation analysis) were performed using IBM^®^ SPSS^®^ Statistics Software (Version 27.0; IBM Corp©, Armonk, NY, USA). In all cases, *p* < 0.05 was set as significant.

## 5. Conclusions

All the analysed plant samples are good sources of polyphenols. The types of phenolic compounds in the plant extracts affected antioxidant activity. Wild-growing plants from Latvia, such as *Q. robur* bark and *C. vulgaris* herbs, had good antioxidant potential; however, *A. absinthium* and *A. vulgaris* herbs had low antioxidant activity in comparison. The types of phenolic compounds present in these plant extracts have a significant impact on antioxidant activity. The plants contained comparable amounts of tannins, which are not often reported in studies but have pharmacological activities. When compared using extraction solvents, all acetone extracts had greater TPC and TFC and better antioxidant activity. In future applications of herbal extracts, it is important to choose an appropriate extraction process for each herb, as the solvent’s type and concentration may change outcomes.

## Figures and Tables

**Figure 1 plants-12-04108-f001:**
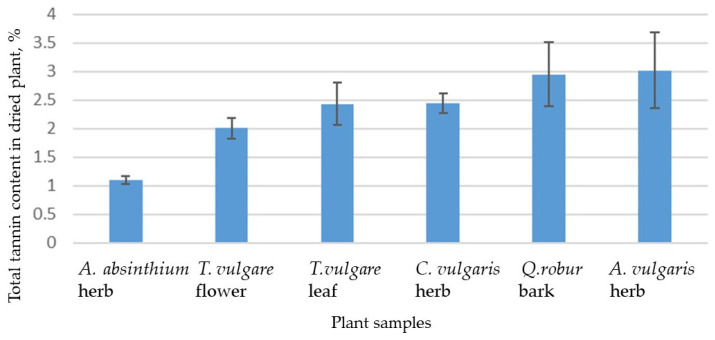
Total tannin content in dried plant samples, % (±SD) of dried mass. *Tanacetum vulgare* L. (*T. vulgare*), *Artemisia absinthium* L. (*A. absinthium*), *Artemisia vulgaris* L. (*A. vulgaris*), *Calluna vulgaris* (L.) *Hull* (*C. vulgaris*), and *Quercus robur* L. (*Q. robur*).

**Figure 2 plants-12-04108-f002:**
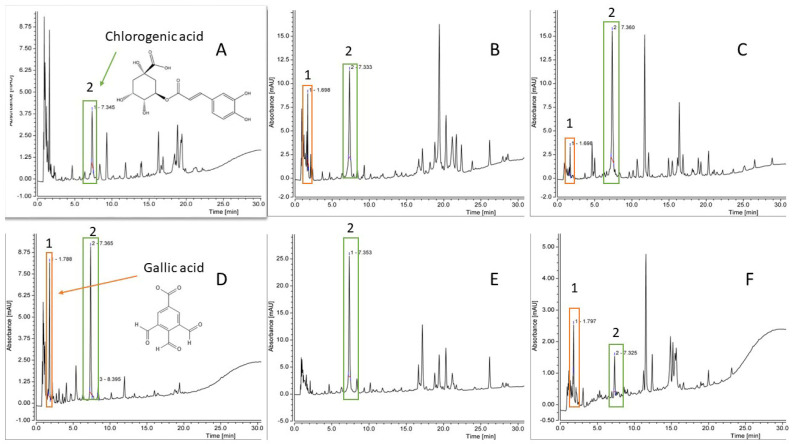
Phenolic compound characterisation via high-performance liquid chromatography (HPLC). (**A**)—*Artemisia vulgaris* herb; (**B**)—*Tanacetum vulgare* flower; (**C**)—*Calluna vulgaris* herb; (**D**)—*Artemisia absinthium* herb; (**E**)—*Tanacetum vulgare* leaf; (**F**)—*Quercus robur* bark. 1—gallic acid; 2—chlorogenic acid.

**Table 1 plants-12-04108-t001:** Total phenolic content, total flavonoid content, and radical scavenging activity of plant extracts.

Plant Sample	TPC (mg GAE/g of Lyophilized Extract wt), ±SD	TFC (mg QE/g of Lyophilized Extract wt), ±SD	IC_50_ Value of DPPH Radical Scavenging Activity (µg/mL), ±SD
Ascorbic acid	-	-	43.92 ± 1.15
*Artemisia absinthium* herb ^a^	68.22 ± 1.62	3.79 ± 0.43	509.33 ± 1.11
*Artemisia vulgaris* herb ^a^	125.04 ± 6.06	15.25 ± 0.51	195.43 ± 1.12
*Calluna vulgaris* herb ^a^	294.88 ± 14.20	51.13 ± 0.29	127.06 ± 1.07
*Quercus robur* bark ^a^	301.39 ± 10.17	5.11 ± 0.32	96.16 ± 1.03
*Tanacetum vulgare* flower ^a^	154.11 ± 7.95	25.12 ± 2.53	193.64 ± 1.10
*Tanacetum vulgare* leaf ^a^	158.48 ± 15.57	46.15 ± 0.29	185.35 ± 1.12
*Artemisia absinthium* herb ^b^	91.12 ± 1.09	5.26 ± 0.99	322.85 ± 1.13
*Artemisia vulgaris* herb ^b^	179.21 ± 2.29	16.06 ± 0.53	164.44 ± 1.13
*Calluna vulgaris* herb ^b^	285.61 ± 5.41	55.08 ± 2.23	104.71 ± 1.07
*Quercus robur* bark ^b^	316.02 ± 21.54	6.20 ± 0.22	83.95 ± 1.04
*Tanacetum vulgare* flower ^b^	155.38 ± 3.17	29.69 ±0.02	181.97 ± 1.07
*Tanacetum vulgare* leaf ^b^	225.99 ± 3.69	52.75 ± 2.37	146.55 ± 1.05

TPC: total phenolic content; TFC: total flavonoid content; GAE: gallic acid equivalents; QE: quercetin equivalents; wt: wight; DPPH: DPPH free radical scavenging activity; ^a^—extract in ethanol; ^b^—extract in acetone.

## Data Availability

Data are contained within the article.

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
