# Peer review of "The Antioxidant Activity of Wild-Growing Plants Containing Phenolic Compounds in Latvia"

_plants, 2023, doi:10.3390/plants12244108_

Round 1

Reviewer 1 Report

Comments and Suggestions for Authors

The antioxidant capacity of herbs is strongly related to their polyphenol content. However, the antioxidant activity of polyphenols is not equal, certain polyphenols exert higher free radical scavenger activity. The identification and quantification of these bioactive molecules is essential considering their potential future use as therapeutic agents. In this research work the antioxidant activity, phenolic and flavonoid content of five different plants native to Latvia were measured. According to the reported data, the polyphenolic composition and the type of extraction solvent have a significant impact on the antioxidant capacity of the samples. However, a few questions and remarks arose while reading the manuscript.
English editing is highly recommended.
Abbreviations must be defined and used after the first appearance.
Line 60-62: “It is odourless, cream-colored needles (from pyridine) or yellow powder, that is slightly solvable in water, ethanol, alkalis but insolvable in ether [15].” Grammatically incorrect sentence. In addition, ellagic acid is soluble in alkalies.
Table 1.: Mistype in the explanation of abbreviations.
Line 122-123: “Gallic acid and chlorogenic acids were found in the C. vulgaris herb, the Q. robur bark, and the A. absinthium herb, but the A. vulgaris herb only presented chlorogenic acid.” In contrast, the presence of gallic acid was indicated on the chromatogram of A. vulgaris.
Figure 2.: In this size and quality this figure is not informative, it would be better to show chromatograms as supplements. Furthermore, the peaks of the tannins on the HPLC chromatograms cannot be indicated without using any standard solutions during the measurements.
Line 136-140: “For the rest of the extracts, radical scavenging activity is as fallow from the lowest: A. absinthium herb extracts < A. vulgaris ethanol extract < T. vulgare flower ethanol extract < T. vulgare leaf ethanol extract < T. vulgare flower acetone extract < A. absinthium acetone extract < T. vulgare leaf acetone extract < C. vulgaris herb extracts.” This sentence is grammatically incorrect. The type of the extracts should be indicated in each case, furthermore, C. vulgaris ethanol extract was not listed.
Line 141-144: “When compared by extraction solvents, all acetone extracts had more TPC, TFC and better antioxidant activity (DPPH and TFC r = -0.319 p = 0.330 DPPH and TPC, r=-0.986, p= 0.000 p<0.01; both acetone and ethanol extracts TPC and DPPH R=1.00, p<0.01, TFC and DPPH no correlation, R= -0.371 p=0.468 acetone, R= -0.429 p = 0.37 ethanol).” The interpretation of the statistical analysis of the samples is not clear.
Line 167: The meaning of DM was not given in the text.
Line 218 and 219: “The 70% ethanol of A. vulgaris had DPPH IC50 0.976 μg/mL, which is higher than our results [51].” Misleading sentence, since the IC50 value of A. vulgaris was lower in the reference research, indicating its higher antioxidant capacity.
Not all of the references meet with the MDPI reference style.
I suggest submitting the manuscript for English editing and minor revision.

Comments on the Quality of English Language

English editing is highly recommended.

Author Response

Dear reviewer, 

We appreciate your comments and suggestions. We have addressed all issues.  Please see the file attached.

Reviewer 2 Report

Comments and Suggestions for Authors

The authors present the Antioxidant Activity of Some Wild-Growing Plants Containing Phenolic Compounds in Latvia.

Overall, the data presented are insufficient for a comprehensive analysis of the antioxidant properties of the six plant species selected in the study.

Thus,

1. What are the reasons for choosing the six plant species and not other species - this aspect could contribute to the improvement of the literature data in the paper.

2. It is also necessary to create a relationship between the chemical species chosen for the chemical quantitative determinations and their expected/desired biogical activities.

3. DPPH method has a very low accuracy; it can be added to other studies, but it cannot be the main method for characterizing antioxidant and biological performances of a series of test extracts.

In conclusion, the paper need major revision.

Author Response

Dear Reviewer, 

we appreciate all your comments and have addressed them to the best ability. Please see the file attached with our detailed response.

Round 2

Reviewer 2 Report

Comments and Suggestions for Authors

>The authors have added new data from the specialized literature to better support the selection of the plant materials, and also made extensive proofreading of the text. Similarly, although the DPPH method is, from my practice, little sensitive by comparison with other chemical methods, I agree that it is perfectly reproducible, cheap and fast, so it has clear advantages. Therefore, please add the explanation of its feasibility (as you described) to argue the benefits of its use in estimating the free radical scavenger potential of the plant extracts.

>On the other hand, I consider necessary new literature data regarding the correlation between the presence/content of tannins in plants and the effect on the microbiome in humans and farm animals, as general data, and, if possible, as particular data toward the plant extracts selected in the present study.

Author Response

Dear reviewer, 

We have revised our manuscript and based on your suggestions added a literature review on DPPH assay and tannin effects on microbiome and health in general. Please see comments below:

>The authors have added new data from the specialized literature to better support the selection of the plant materials, and also made extensive proofreading of the text. Similarly, although the DPPH method is, from my practice, little sensitive by comparison with other chemical methods, I agree that it is perfectly reproducible, cheap and fast, so it has clear advantages. Therefore, please add the explanation of its feasibility (as you described) to argue the benefits of its use in estimating the free radical scavenger potential of the plant extracts.

ANSWER:

Thank you for your suggestions, we think that discussing the benefits of the DPPH method gives better justification for a selection of assay.

We have added lines 280-290 in the discussion section.

The DPPH radical scavenging assay is a commonly used method to measure antioxidant efficiency, although it has limits in sensitivity. However, this assay has benefits such as accuracy, reproducibility, and simplicity, and it is an economically feasible method [58] The DPPH radical scavenging assay is not the only method to determine the potential antioxidant activity of herbal material. Various assays can be applied to measure antioxidant activities such as spectrometric (TRAP—Total Peroxyl Radical Trapping Antioxidant Parameter; CUPRAC—Cupric Reducing Antioxidant Power; FRAP—Ferric Reducing Antioxidant Power; PFRAP—potassium ferricyanide reducing power etc.), chromatographic (gas or high-performance liquid), and Electrochemical techniques [59]. It should be noted that the combination of several methods is advantageous as it gives a better understanding of extracts antioxidant ability.

>On the other hand, I consider necessary new literature data regarding the correlation between the presence/content of tannins in plants and the effect on the microbiome in humans and farm animals, as general data, and, if possible, as particular data toward the plant extracts selected in the present study.

ANSWER:

Thank you for this comment. We added some clarification on importance of tannin level monitoring herbal medicine and animal feed supplementation. We think that this will give better understanding about tannins’ impact on health and same risks related to high intake of tannin rich products.

The discussion section lines: 291-308

Tannins have been considered antinutritional for many years, but recent studies show their health benefits. To evaluate tannin’s effect on health several factors must be considered such as animal species, types of tannins and concentrations in plants or feed [60,61]. A review study has reported that tannins have antibacterial effects, improve the intestinal microbial ecosystem, and enhance the growth and well-being of poultry and monogastric animals if used in small levels (<5% of dry weight of feed)[61]. Some animals, including humans, pigs, and mice, have proline-rich enzymes in their saliva that neutralize negative effects of tannins [60,62]. In addition to the levels of tannins in feeds and herbs, type of tannins influences health benefits. Hydrolysable tannins did not show a negative impact on animal well-being, although high doses can cause adverse effects. Condensed tannins are considered safe in moderate amounts [60,61,63]. A study of the human gut microbiome showed positive effects of condensed tannins [62]. Ruminants can benefit from tannin intake as it reduces bloating and intestinal parasites [60,64]. Tannin metabolism and its relationship with gut microbiome is complex due to their diversity and high molecular mass. Hydrolysable tannins interact differently with gut microbiome than condensed tannins, and often tannin metabolites are responsible for systemic pharmacological effects [65]. Although our study’s plants have reports on various polyphenols, including tannins, relation to antibacterial, antiparasitic, antioxidant, and anti-inflammatory effects, there are still gaps in knowledge about precise effects and correlation of compounds and activities. Therefore, it is necessary to monitor tannin levels in herbal medicine and determine their effects on health.